# Mortality in children under 5 years of age with congenital syphilis in Brazil: A nationwide cohort study

Enny S. Paixao[1,2]*, Andrêa JF Ferreira[2], Idália Oliveira dos Santos[2], Laura C. Rodrigues[1,2], Rosemeire Fiaccone[2,3], Leonardo Salvi[2], Guilherme Lopes de Oliveira[2,4], José Guilherme Santana[2,3], Andrey Moreira Cardoso[5], Carlos Antônio de S. S. Teles[2], Maria Auxiliadora Soares[2], Eliana Amaral[6], Liam Smeeth[1], Mauricio L. Barreto[2,7], Maria Yury Ichihara[2]

1 London School of Hygiene and Tropical Medicine, London, United Kingdom, 2 Centro de Integração de Dados e Conhecimentos para Saúde (CIDACS), FIOCRUZ, Salvador, Bahia, Brasil, 3 Instituto de Matemática e Estatística, Universidade Federal da Bahia (UFBA), Salvador, Bahia, Brasil, 4 Centro Federal de Educação Tecnológica de Minas Gerais (CEFET-MG), Belo Horizonte, Minas Gerais, Brasil, 5 Escola Nacional de Saúde Pública (ENSP), FIOCUZ, Rio de Janeiro, Rio de Janeiro, Brasil, 6 School of Medical Sciences, University of Campinas (UNICAMP), Campinas, Sao Paulo, Brasil, 7 Instituto de Saúde Coletiva, Universidade Federal da Bahia (UFBA), Salvador, Bahia, Brasil

* Enny.cruz@lshtm.ac.uk

**Data Availability Statement:** All data supporting the findings presented here were obtained from Centro de Integração de Dados e Conhecimentos para Saúde (CIDACS). Importantly, restrictions

## Abstract

### Background

Congenital syphilis (CS) is a major and avoidable cause of neonatal death worldwide. In this study, we aimed to estimate excess all-cause mortality in children under 5 years with CS compared to those without CS.

### Methods and findings

In this population-based cohort study, we used linked, routinely collected data from Brazil from January 2011 to December 2017. Cox survival models were adjusted for maternal region of residence, maternal age, education, material status, self-declared race and newborn sex, and year of birth and stratified according to maternal treatment status, non-treponemal titers and presence of signs and symptoms at birth. Over 7 years, a total of 20 057 013 live-born children followed up (through linkage) to 5 years of age, 93 525 were registered with CS, and 2 476 died. The all-cause mortality rate in the CS group was 7·84/1 000 person-years compared with 2·92/1 000 person-years in children without CS, crude hazard ratio (HR) = 2·41 (95% CI 2·31 to 2·50). In the fully adjusted model, the highest under-five mortality risk was observed among children with CS from untreated mothers HR = 2·82 (95% CI 2·63 to 3·02), infants with non-treponemal titer higher than 1:64 HR = 8·87 (95% CI 7·70 to 10·22), and children with signs and symptoms at birth HR = 7·10 (95% CI 6·60 to 7·63). Among children registered with CS, CS was recorded as the underlying cause of death in 33% (495/1 496) of neonatal, 11% (85/770) of postneonatal, and 2·9% (6/210) of children 1 year of age. The main limitations of this study were the use of a secondary

apply to the availability of these data. However, upon reasonable request and provided all ethical and legal requirements are met, the institutional data curation team can make the data available. Information on how to apply to access the data can be found at https://cidacs.bahia.fiocruz.br/en/.

**Funding:** This work was supported by the National Council for Scientific and Technological Development CNPq/Brazil (442873/2019-0 to MYI); the Bill and Melinda Gates Foundation (401739/2015-5 to MB) and Wellcome Trust (202912/Z/16/Z to MB and 213589/Z/18/Z to ESP). The funders had no role in study design, data collection and analysis, decision to publish, or preparation of the manuscript.

**Competing interests:** The authors have declared that no competing interests exist.

**Abbreviations:** CI, confidence interval; CIDACS, Center of Data and Knowledge Integration for Health; CS, congenital syphilis; HR, hazard ratio; ICD, International Classification of Diseases; PY, person-year; SIM, Mortality Information System.

database without additional clinical information and the potential misclassification of exposure status.

## Conclusions

This study showed an increased mortality risk among children with CS that goes beyond the first year of life. It also reinforces the importance of maternal treatment that infant non-treponemal titers and the presence of signs and symptoms of CS at birth are strongly associated with subsequent mortality.

## Trial registration

Observational study.

---

## Author summary

### Why was this study done?

- Despite global initiatives to eliminate mother-to-child transmission of syphilis and congenital syphilis, analyses of national data from several countries have shown an alarming rise in the number of congenital syphilis (CS) cases.

- The number of infants dying annually from congenital syphilis is unclear, but estimates suggest more than 661 000 CS cases would occur worldwide.

- There is a lack of studies estimating the excess all-cause mortality among live-born children with CS compared with live-born children without CS and with follow-up after the first year of life.

### What did the researchers do and find?

- We analyzed a historical cohort of more than 20 million births in Brazil from 2011 to 2017.

- Live births with CS were 2 times more likely to die than their counterparts without CS, with the increased risk not totally explained by prematurity and low birth weight.

- The greatest mortality risks were observed in children with CS born to untreated mothers, with higher antibody titers, and with signs and symptoms of CS at birth.

- In live-born children registered with CS registry, CS and CS-related causes of death mainly drove the excess mortality, and specific external causes of death among older children with CS were more frequent than among those without CS.

### What do these findings mean?

- The evidence highlights the importance of ensuring public health action for timely detection and maternal treatment to reduce vertical transmission and mitigate adverse pregnancy outcomes and CS-related child mortality.

## Introduction

Congenital syphilis (CS) is caused by the transmission of the spirochete *Treponema pallidum* from mother to fetus, resulting in a multiorgan infection that may cause neurologic and musculoskeletal disabilities [1–3]. CS is preventable with maternal screening and timely and adequate treatment during pregnancy [1]. Despite global initiatives to eliminate mother-to-child transmission of syphilis and CS, analyses of national data from several countries, including the United States of America, Canada, and Brazil, have shown an alarming rise in the number of CS cases [4–7].

CS is considered an antenatal care failure since diagnosis and treatment for pregnant women is well known, not expensive, and given at least 28 days before delivery can prevent the disease in the offspring. However, from 2014 to 2016, several pharmaceutical manufacturers stopped producing penicillin, which dramatically increased the stock-out risk and 38 countries, experienced a severe penicillin supply shortage, resulting in a lack of correct treatment among pregnant women and infants with syphilis [8,9].

Due to difficulty in diagnosis and lack of standardized globally accepted case definition, it is challenging to estimate the burden of worldwide CS accurately [10]. However, data from 2016 suggest that more than 661 000 CS cases would occur worldwide [10]. Although the diagnosis of syphilis in pregnant women is straightforward, the diagnosis of CS is complex because most infants are asymptomatic at birth or have unspecific signs and symptoms [11], and no reliable laboratory test identifies asymptomatic CS at birth [12]. Therefore, clinicians rely on reviewing maternal and neonatal serology, maternal treatment history, and follow-up of the exposed infants to make a presumptive diagnosis [3].

The number of infants dying annually from CS is unclear, but estimates suggest over 370 000 perinatal deaths and 21 500 additional CS-related deaths [13,14] by the first year of life due to complications related to prematurity, low birth weight, and congenital infections [14]. Nevertheless, the relative risk of death among live-born children with CS, as compared with those without the conditions, the role of important predictors of CS, such as maternal treatment and the presence of signs and symptoms at birth, remains unclear.

Leveraging national population-based, linked data on more than 20 million live-born children between 2011 and 2017 in Brazil, a country with a considerable burden of CS in the Americas, we aimed to estimate excess all-cause mortality in children under 5 years with CS as compared to those without CS. Furthermore, we compare mortality rates among live-born children with CS, according to maternal treatment status, non-treponemal titer, and the presence of signs and symptoms at birth, with the rate among children without CS. Additionally, we examined the underlying causes of death among children born with CS.

## Methods

### Study design

We conducted a retrospective population-based cohort study, including all singleton live births in Brazil from January 1, 2011 to December 31, 2017. These live births were followed up from birth until December 31, 2017, death, or up to 5 years old. The follow-up was designed using the record linkage strategy by liking birth with death information.

A written analysis plan was not developed; however, all analyses were presented and discussed during regular study group meetings. This study is reported as per the Reporting of studies Conducted using Observational Routinely collected health Data (RECORD) guideline (S1 Table).

## Data source

This study used routine data from the Brazilian Information System. To identify births in Brazil, we used data from the Live Births Information System-SINASC [15]. The SINASC, an information system with 99.8% coverage in Brazil [16], records data from the Declaration of Live Births, a legal document completed by the health worker who assists in the delivery. From this system, we obtained information about the mother (maternal age, education, marital status, and ethnicity), the pregnancy (number of prenatal appointments and gestational age at birth), the newborn (birth weight and sex), and pregnancy history (previous stillbirth). Data completeness is very high; most variables were >90% complete [17].

Death-related information was obtained from the Mortality Information System (SIM), which records death certificates. The death certificate is a legal document that must be completed by a physician who attests to the cause of death. We obtained information on the date and the underlying cause of death according to the International Classification of Diseases/ICD 10. As of 2011, SIM registered 96.1% of deaths in Brazil [18].

We obtained information about CS from the Information System for Notifiable Disease (SINAN-Syphilis). In Brazil, the registration of suspected CS cases is compulsory. From SINAN-Syphilis, we retained data on final disease classification (confirmed or excluded), newborn laboratory tests and symptoms, and treatment information for both mothers and newborns.

## Linkage process

Live births records from SINASC were linked with SINAN and SIM. Name, date of birth or age, and residence of the mother were used as matching variables. The linkage was performed with CIDACS-RL-Record Linkage, a novel record-linkage tool developed to link large-scale administrative datasets at Center of Data and Knowledge Integration for Health (CIDACS), applying the combination of indexing and searching algorithms approach [19] to search the most similar records from the Indexed SIM and SINAN-Syphilis for each record in SINASC and submit them to the pairwise comparisons step. Candidate linking records were ordered by the scores, and only the comparison pair with the highest score was retained as a potential link. All remaining candidate records were discarded. Barbosa and colleagues detailed the record linkage process [19]. For the linkage between SINASC and SIM, a sample of 2,000 pairs stratified in 3 categories of linkage score (high score—above 0·95, intermediate score—values between 0·90 and 0·95, and low score—below 0·90) was obtained and manually reviewed to evaluate the linkage quality. We obtained a mean sensitivity and specificity of over 93% in this validation process. For the linkage SINASC-SINAN-Syphilis, we compared the number of individuals registered with confirmed CS in the national statistics and compared it with the number of matches in our linked dataset. In this validation process, we obtained a sensitivity of about 85.3%. All identifiable data were extracted in 2020 and made available by the Brazilian Ministry of Health. Linkage procedures were conducted at CIDACS in a strict data protection environment and according to ethical and legal rules [20].

## Procedures

We included all singleton live births who contributed records during the study period. We excluded records registered in SINAN-Syphilis suspected of CS but ruled out after epidemiological investigation and those classified as fetal loss that occurred before 22 weeks of gestation.

In Brazil, live births who meet 1 or more of the following criteria should be reported and investigated as suspected cases of congenital syphilis: (1) Live births from mothers with untreated or inadequately treated syphilis; (2) Children under 13 years of age with at least 1 of the following situations: (a) clinical, cerebrospinal fluid, or radiological manifestation of CS

and a reactive non-treponemal test; (b) infants with non-treponemal test titers greater than the maternal titers in at least 2 dilutions of peripheral blood samples; (c) children with ascending non-treponemal test titers in at least 2 dilutions; (d) titers of non-treponemal tests still positive after 6 months of age, in a child adequately treated in the neonatal period; (e) positive treponemal test at 18 months of age, without previous diagnose of CS; and (3) microbiological evidence of *Treponema pallidum* in a nasal discharge sample or skin lesion, tissue biopsy, or autopsy.

In this study, we considered CS case live births classified in SINAN-Syphilis as confirmed cases and their register was linked with a SINASC record. We then classified CS cases according to maternal treatment status [adequate (treatment carried out in accordance with the recommended scheme for the maternal clinical phase of syphilis, started more than 30 days before childbirth and whose titers dropped as expected and remained low), inadequate, and non-treated], newborn non-treponemal titer (<1:16, 1:32, 1:64) and live-born children with CS into 2 categories: those with signs and symptoms (recorded signs and symptoms: jaundice, anemia, splenomegaly, osteochondritis, blood mucus rhinitis, and hepatomegaly) and those without signs and symptoms.

We classified mortality as neonatal mortality (from birth up to 27 days), postneonatal (28 to 364 days), mortality between 1 to 4 years (after the first year of life up to 5 years, the events in the first year were excluded), and under-five mortality (from birth up to 5 years).

## Statistical analyses

Descriptive statistics are presented for maternal sociodemographic data and newborn characteristics. Mortality rates (deaths/1 000-person-year (PY) and crude hazard ratios (HRs) with 95% confidence intervals (CIs) comparing live births with CS to live births without CS were estimated using Cox proportional hazards models. We used age attained within the study for the time scale in our survival analyses. The covariates included in the adjustments were selected a priori based on the literature. In the partially adjusted model, we included region and year of birth. In the fully adjusted model, we added maternal age, education, material status, self-declared race, and newborn sex. We also examined the association of newborn maternal treatment status, newborn non-treponemal titer, and the presence of symptoms at birth. A sensitivity analysis of HR was planned a priori, with the cohort restricted to individuals with gestational age at birth <37 weeks and birth weight more than 2 500 g to observe the excess mortality independent of these conditions.

We assessed the proportional mortality (number of deaths by specific causes/total number of deaths) of the underlying cause of death using ICD-10 codes. Then, the proportional mortality ratio by cause of death was calculated to indicate the chance of children born with CS dying from a given cause compared to children without CS. We did post hoc sensitivity analyses further describing deaths coded as CS in the mortality system but without a registry in SINAN-Syphilis.

This study analyzed de-identified data and was approved by the Research Ethics Committee of the Federal University of Bahia Institute of Health Collective Research Ethics Committee (CAAE registration no. 18022319.4.0000.5030). No informed consent was required for this study.

Data analyses were performed in Stata version 15·0.

## Results

We followed 20 057 013 live-born children from birth up to 5 years (mean, 46 months; range, 0 to 59) (S1 Fig). The characteristics of the study population are reported in Table 1. In general,

**Table 1. Baseline characteristics of 20,057,013 singleton live births in the cohort-linked data by CS status.**

| | Live births without CS (no SINAN registry-19,963,488) n (%) | Live births with CS (registered in SINAN-93,525) n (%) |
|---|---|---|
| **Maternal age (years)** | | |
| <20 | 3,701,410 (18.54) | 23,402 (25.02) |
| 20–34 | 13,809,529 (69.18) | 62,277 (66.59) |
| 35+ | 2,452,059 (12.28) | 7,841 (8.38) |
| Missing | 490 | 5 |
| **Marital status** | | |
| Single | 8,458,975 (42.89) | 64,948 (70.14) |
| Widow | 36,858 (0.19) | 184 (0.20) |
| Divorced | 213,757 (1.08) | 722 (0.78) |
| Married/union | 11,011,301 (55.84) | 26,743 (28.88) |
| Missing | 242,597 | 928 |
| **Maternal education** | | |
| none | 134,053 (0.68) | 1,094 (1.19) |
| 1–3 years | 690,120 (3.52) | 5,491 (5.96) |
| 4–7 years | 4,018,554 (20.49) | 34,748 (37.69) |
| 8–12 years | 11,315,310 (57.68) | 48,474 (52.57) |
| 12+ years | 3,458,004 (17.63) | 2,393 (2.60) |
| Missing | 347,447 | 1,325 |
| **Maternal ethnicity** | | |
| White | 6,513,133 (36.57) | 19,611 (22.58) |
| Black | 975,478 (5.48) | 9,632 (11.09) |
| Asian | 70,376 (0.40) | 228 (0.26) |
| Mixed brown | 10,104,490 (56.73) | 57,037 (65.67) |
| Indigenous | 147,281 (0.83) | 348 (0.40) |
| Missing | 2,152,730 | 6,669 |
| **Previous stillbirth** | | |
| Yes | 3,259,661 (17.99) | 22,375 (25.61) |
| No | 14,864,332 (82.01) | 64,989 (74.39) |
| Missing | 1,839,495 | 6,161 |
| **Number of prenatal appointments** | | |
| None | 483,261 (2.42) | 7,682 (8.22) |
| 0–3 appointments | 1,329,985 (6.67) | 14,942 (15.99) |
| 4–6 appointments | 5,038,084 (25.25) | 29,059 (31.09) |
| 7+ appointments | 13,099,976 (65.66) | 41,777 (44.70) |
| Missing | 12,182 | 65 |
| **Mode of delivery** | | |
| Vaginal | 8,913,251 (44.72) | 59,062 (63.22) |
| C-section | 11,018,115 (55.28) | 34,360 (36.78) |
| Missing | 32,122 | 103 |
| **Year of birth** | | |
| 2011 | 2,840,871 (14.23) | 7,375 (7.89) |
| 2012 | 2,842,332 (14.24) | 9,030 (9.66) |
| 2013 | 2,832,111 (14.19) | 10,985 (11.75) |
| 2014 | 2,902,694 (14.54) | 13,009 (13.91) |
| 2015 | 2,931,845 (14.69) | 15,653 (16.74) |

*(Continued)*

**Table 1.** (Continued)

| | Live births without CS (no SINAN registry-19,963,488) *n* (%) | Live births with CS (registered in SINAN-93,525) *n* (%) |
|---|---|---|
| 2016 | 2,778,195 (13.92) | 17,261 (18.46) |
| 2017 | 2,835,440 (14.20) | 20,212 (21.61) |
| Missing | - | - |
| **Birth region** | | |
| N | 2,153,414 (10.79) | 7,576 (8.10) |
| NE | 5,658,627 (28.35) | 27,551 (29.46) |
| SE | 7,865,186 (39.40) | 40,667 (43.48) |
| S | 2,662,339 (13.34) | 12,484 (13.35) |
| MW | 1,621,717 (8.12) | 5,244 (5.61) |
| Missing | 2,205 | 3 |
| **Sex of the newborn** | | |
| Female | 9,729,472 (48.74) | 46,466 (49.70) |
| Male | 10,230,501 (51.26) | 47,032 (50.30) |
| Missing | 3,515 | 27 |
| **Gestational age at birth (weeks)** | | |
| <32 | 256,393 (1.43) | 2,841 (3.25) |
| 32–36 | 1,671,018 (9.29) | 12,314 (14.07) |
| 37+ | 16,058,540 (89.28) | 72,385 (82.69) |
| Missing | 1,977,537 | 5,985 |
| **Birth weight (g)** | | |
| <1,500 | 225,126 (1.13) | 2,374 (2.54) |
| 1,500–2,499 | 1,235,548 (6.19) | 13,699 (14.65) |
| 2,500+ | 18,486,179 (92.68) | 77,410 (82.81) |
| Missing | 16,635 | 42 |
| **Small for gestational age** | | |
| Yes | 1,310,370 (7.38) | 11,344 (13.13) |
| No | 16,440,496 (92.62) | 75,068 (86.87) |
| Missing | 2,212,622 | 7,113 |
| **Apgar score at 5 min** | | |
| <7 | 214,206 (1.10) | 2,018 (2.22) |
| 7–10 | 19,195,358 (98.90) | 89,047 (97.78) |
| Missing | 553,924 | 2,460 |
| **Mortality** | | |
| Neonatal | 137,020 (0.69) | 1,496 (1.62) |
| Postneonatal | 59,974 (0.30) | 770 (0.84) |
| 1–4 years | 26,489 (0.13) | 210 (0.23) |

CS, congenital syphilis.

the 93 525 live-born children registered with CS had a higher proportion of younger, single black and mixed brown mothers with fewer years of education than the 19 963 488 children without CS registry. A total of 17·3% (15 155/87 540) of live birth children with CS were born preterm, 17·2% had low birth weight (16 073/93 483), and 13·1% (11 344/86 412) were small for gestational age compared with 10·7% (1 927 411/17 985 951), 7·3% (1 460 674/19 946 853), and 7·4% (1 310 370/17 750 866) among non-CS cases, respectively (Table 1).

Of 93 525 live-born children registered with CS, 88 965 had information on live birth treatment. Of those, nearly 95% (84 302/88 965) of the cases were treated and 56·89% (50 613/88 965) received aqueous crystalline penicillin, 12·41% (11 037/88 965) received procaine penicillin, and 7·49% (6 660/88 965) received benzathine penicillin. Almost 18% (15 992/88 965) received another treatment scheme. More than 71% (369/518) of infants who died untreated died within the first 24 h of life, increasing to 87·06% (451/518) by the end of the first week. Regarding maternal treatment, complete information was available for 83 098 live births children with CS, 4·62% (3 835/83 098) of mothers were adequately treated, 65·59% (54 502/83 098) were inadequately treated, and 29·80% (24 761/83 098) did not receive treatment during pregnancy. Around 78% (73 345/93 525) of live-born children registered with CS had results for a non-treponemal test. About 10% (9 722/93 525) had symptoms recorded; the most common symptom was jaundice, followed by hepatomegaly and anemia.

During follow-up, 2 476 registered children with CS died, and the all-cause mortality rate in this group was 7·84 per 1 000 person-years compared with 2·92 per 1 000 person-years in children without CS. The crude under-five mortality rate among live-born children with CS was 2·41 times (95% CI 2·31 to 2·50), as high as that among live-born children without CS. In the fully adjusted model, the under-five mortality was 2 times higher among those with CS (95% CI 1·97 to 2·15) than their counterparts without CS. In the analyses stratified by age, during the neonatal period, the fully adjusted model returned HR = 2·07 (95% CI 1·96 to 2·19), and during the postneonatal period, we had HR = 2·22 (95% CI 2 06 to 2 40) and among children, 1 year of age or older, we found HR = 1·56 (95% CI 1·36 to 1·80) (Table 2).

Maternal treatment status, offspring's non-treponemal titers, and symptoms' presence significantly affected the mortality of children with CS. Regarding maternal treatment, we observed a dose-response association, with the lowest under-five mortality risk detected among children from adequately treated mothers [HR 1·54 (95% CI 1·21 to 1·96)] and the highest under-five mortality risk among children with CS from untreated mothers [HR 2·82 (95% CI 2·63 to 3·02)] compared to children without CS (Table 3). This pattern was observed in all age groups, except among children 1 year of age or older. In the sensitivity analysis restricted to children born at term and with normal birth weight, live-born children registered with CS were 1 51 (95% CI 1·39 to 1·64) more likely to die than their counterparts without CS (Table 3).

The higher the infant non-treponemal titers, the higher the mortality risk, except for those older than 1 year. During the neonatal period, the hazard among those with a titers higher than 1:64 was 9·35 (95% CI 7 84 to 11 15) times higher than among those without a CS registry (Table 4). For the presence of symptoms, in the fully adjusted model, the under-five HR was 7·10 (95% CI 6·60 to 7·63) among children with symptomatic CS and 1·52 (95% CI 1·45 to 1·60) among children with CS, but for whom no symptoms were recorded compared to children without CS (Table 5).

The most common underlying causes of death among live-born children reported with CS by age are described in Fig 1. Among live-born children registered with CS, this condition was reported in 33·09% of the neonatal, 11·04% of postneonatal, and 2·86% of children 1 year of age or older underlying cause of death. On the other hand, 730 (0.3%) deaths in the group without a SINAN-registry of CS had CS coded as the underlying cause of death (464 as the underlying cause and 266 in another component of the chain of events leading to death). During the neonatal period, perinatal complications were the most frequent causes of death registered (mainly maternal factors and complications of pregnancy; disorders related to short gestation and low birth weight, asphyxia, and other respiratory distress and sepsis). For the postneonatal period and among older children, pneumonia, sepsis, congenital malformations of the heart, and unknown causes of death appeared as the most common causes.

We conducted a post hoc analysis describing the deaths in the group without a SINAN registry of CS who had CS as the underlying cause of death (S2 Table). The 2 groups of children

**Table 2. Mortality risk by age group among singleton live births in the cohort-linked data, Brazil, 2011–2017.**

| Live births | Number of deaths | Person-years | Mortality rate per 1 000 person-years | Crude model HR (95% CI) | Partly adjusted model HR (95% CI)* | Fully adjusted model HR (95% CI)† |
|---|---|---|---|---|---|---|
| **Overall population of live born** | | | | | | |
| **Neonatal mortality (under 28 days)** | | | | | | |
| Without CS | 137 020 | 1 467 358 | 93·37 | Ref | Ref | Ref |
| CS‡ | 1 496 | 6 824 | 219·21 | 2·33 (2·22–2·46) | 2·33 (2·22–2·46) | 2·07 (1·96–2·19) |
| **Postneonatal mortality (28–364 days)** | | | | | | |
| Without CS | 59 974 | 18 253 627 | 3·28 | Ref | Ref | Ref |
| CS‡ | 770 | 84 384 | 9·12 | 2·77 (2·58–2·97) | 2·77 (2·58–2·98) | 2·22 (2·06–2·40) |
| **Mortality between 1–4 years** | | | | | | |
| Without CS | 26 489 | 56 659 192 | 0·46 | Ref | Ref | Ref |
| CS‡ | 210 | 224 372 | 0·93 | 1·92 (1·67–2·20) | 1·85 (1·61–2·12) | 1·56 (1·36–1·80) |
| **Under-five mortality (under 5 years)** | | | | | | |
| Without CS | 223 483 | 76 403 230 | 2·92 | Ref | Ref | Ref |
| CS‡ | 2 476 | 315 654 | 7·84 | 2·41 (2·31–2·50) | 2·40 (2·31–2·50) | 2·06 (1·97–2·15) |
| **Excluding preterm and low birth weight live born** | | | | | | |
| **Neonatal mortality (under 28 days)** | | | | | | |
| Without CS | 30 131 | 1 143 160 | 26·35 | Ref | Ref | Ref |
| CS‡ | 196 | 4 910 | 38·91 | 1·51 (1·31–1·74) | 1·53 (1·33–1·76) | 1·31 (1·13–1·51) |
| **Postneonatal mortality (28–364 days)** | | | | | | |
| Without CS | 28 810 | 14 244 595 | 2·02 | Ref | Ref | Ref |
| CS‡ | 266 | 61 074 | 4·35 | 2·15 (1·90–2·42) | 2·16 (1·92–2·44) | 1·65 (1·46–1·87) |
| **Mortality between 1–4 years** | | | | | | |
| Without CS | 17 770 | 42 599 572 | 0·41 | Ref | Ref | Ref |
| CS‡ | 131 | 155 874 | 0·84 | 1·94 (1·63–2·30) | 1·88 (1·58–2·23) | 1·58 (1·32–1·88) |
| **Under-five mortality (under 5 years)** | | | | | | |
| Without CS | 76 711 | 58 003 198 | 1·32 | Ref | Ref | Ref |
| CS‡ | 593 | 221 906 | 2·67 | 1·84 (1·70–2·00) | 1·85 (1·70–2·00) | 1·51 (1·39–1·64) |

‡Those individuals with SINAN registry.

*Partly adjusted for region and year of birth.

†Adjusted for maternal age, education, marital status, and self-declared race, newborn sex, region, and year of birth.

CI, confidence interval; CS, congenital syphilis; HR, hazard ratio.

with CS (those with SINAN registry and those without SINAN registry) had similar baseline characteristics. However, the median time from birth to death was 3.5 days in the group without a CS registry, coded as death due to CS, and 11.5 days in the group with a CS registry.

## Discussion

Analyses of Brazilian national registry-based data showed that the under-five mortality rates among children with CS were twice as high as those without CS. We observed a dose-response association of maternal treatment status and infant non-treponemal titers with the highest mortality rates observed among children born from untreated women and infants with higher

**Table 3. Mortality risk by age group and maternal treatment status among singleton live births in the cohort-linked data, Brazil, 2011–2017.**

| Live births | Number of deaths | Person-years | Mortality rate per 1 000 person-years | Crude model HR (95% CI) | Partly adjusted model HR (95% CI)* | Fully adjusted model HR (95% CI)† |
|---|---|---|---|---|---|---|
| **Neonatal mortality (under 28 days)** | | | | | | |
| Without CS | 137 020 | 1 467 358 | 93·37 | Ref | Ref | Ref |
| CS‡ mothers adequate treated | 43 | 281 | 152·97 | 1·63 (1·21–2·20) | 1·63 (1·21–2·21) | 1·46 (1·06–2·00) |
| CS‡ mothers inadequate treated | 712 | 3 986 | 178·58 | 1·90 (1·77–2·05) | 1·90 (1·77–2·05) | 1·70 (1·58–1·84) |
| CS‡ mothers not treated | 576 | 1 795 | 320·8 | 3·41 (3·14–3·70) | 3·41 (3·14–3·70) | 3·03 (2·77–3·30) |
| **Postneonatal mortality (28–364 days)** | | | | | | |
| Without CS | 59 974 | 18 253 627 | 3·25 | Ref | Ref | Ref |
| CS‡ mothers adequate treated | 19 | 3 487 | 5·44 | 1·65 (1·05–2·59) | 1·63 (1·04–2·57) | 1·44 (0·91–2·29) |
| CS‡ mothers inadequate treated | 398 | 49 355 | 8·06 | 2·45 (2·22–2·70) | 2·45 (2·22–2·70) | 1·99 (1·79–2·20) |
| CS‡ mothers not treated | 264 | 22 135 | 11·92 | 3·62 (3·21–4·08) | 3·64 (3·23–4·11) | 2·83 (2·48–3·22) |
| **Mortality between 1–4 years** | | | | | | |
| Without CS | 26 489 | 56 659 192 | 0·46 | Ref | Ref | Ref |
| CS‡ mothers adequate treated | 11 | 8 551 | 1·28 | 2·59 (1·43–4·68) | 2·41 (1·33–4·35) | 2·28 (1·26–4·13) |
| CS‡ mothers inadequate treated | 114 | 130 148 | 0·87 | 1·79 (1·49–2·15) | 1·71 (1·42–2·06) | 1·47 (1·21–1·77) |
| CS‡ mothers not treated | 61 | 60 878 | 1·00 | 2·08 (1·61–2·67) | 2·05 (1·59–2·63) | 1·74 (1·34–2·26) |
| **Under-five mortality (under 5 years)** | | | | | | |
| Without CS | 223 483 | 76 403 230 | 2·90 | Ref | Ref | Ref |
| CS‡ mothers adequate treated | 73 | 12 322 | 5·90 | 1·73 (1·38–2·18) | 1·72 (1·36–2·16) | 1·54 (1·21–1·96) |
| CS‡ mothers inadequate treated | 1 224 | 183 532 | 6·60 | 2·04 (1·93–2·16) | 2·03 (1·92–2·15) | 1·76 (1·66–1·86) |
| CS‡ mothers not treated | 901 | 84 830 | 10·6 | 3·32 (3·11–3·55) | 3·32 (3·11–3·55) | 2·82 (2·63–3·02) |

‡Those individuals with SINAN registry.

*Partly adjusted for region and year of birth.

†Adjusted for maternal age, education, marital status, and self-declared race, region, and year of birth.

CI, confidence interval; CS, congenital syphilis; HR, hazard ratio.

test titers. Live born neonates with signs and symptoms of CS recorded at birth had higher mortality rates than their counterparts without symptoms. We observed that among children registered with CS, 33% of neonatal, 11% of postneonatal, and 2·9% of children 1 year of age, had CS recorded as the underlying cause of death. Hematological disorders of newborns, anemia, and bacterial meningitis were underlying causes of death more commonly recorded among children with CS than those without CS.

Previous studies have estimated the mortality risk among infants with CS [13,21–23]. However, to our knowledge, this is the first study with a comparison group and long-term follow-up that revealed that even after the first year of life, children with CS still have higher mortality rates than their counterparts without CS.

As a multiorgan infection, CS can cause several manifestations in the central nervous system, hematological, liver, and spleen that can lead to death [13]. Even among asymptomatic infants at birth, close follow-up is recommended, whether or not treated, because CS

**Table 4. Mortality risk by age group and non-treponemal titers among singleton live births in the cohort-linked data, Brazil, 2011–2017.**

| Live births | Number of deaths | Person-years | Mortality rate per 1 000 person-years | Crude model HR (95% CI) | Partly adjusted model HR (95% CI)* | Fully adjusted model HR (95% CI)† |
|---|---|---|---|---|---|---|
| **Neonatal mortality (under 28 days)** | | | | | | |
| Without CS | 137 020 | 1 467 358 | 93·37 | Ref | Ref | Ref |
| CS‡ ≤1:16 | 507 | 4 813 | 105·32 | 1·12 (1·03–1·23) | 1·13 (1·03–1·23) | 0·99 (0·91–1·09) |
| CS‡ >1:16 ≤1:32 | 116 | 293 | 395·38 | 4·19 (3·49–5·03) | 4·21 (3·51–5·05) | 3·64 (3·00–4·42) |
| CS‡ >1:32 ≤1:64 | 85 | 132 | 641·54 | 6·76 (5·46–8·36) | 6·80 (5·49–8·41) | 6·04 (4·83–7·55) |
| CS‡ >1:64 | 133 | 136 | 977·24 | 10·19 (8·60–12·08) | 10·31 (8·70–12·23) | 9·35 (7·84–11·15) |
| **Postneonatal mortality (28–364 days)** | | | | | | |
| Without CS | 59 974 | 18 253 627 | 3·28 | Ref | Ref | Ref |
| CS‡ ≤1:16 | 433 | 59 657 | 7·25 | 2·20 (2·00–2·42) | 2·21 (2·01–2·43) | 1·79 (1·62–1·98) |
| CS‡ >1:16 ≤1:32 | 56 | 3 598 | 15·56 | 4·72 (3·63–6·13) | 4·73 (3·64–6·15) | 3·58 (2·71–4·74) |
| CS‡ >1:32 ≤1:64 | 42 | 1 604 | 26·17 | 7·93 (5·86–10·73) | 7·94 (5·86–10·75) | 5·59 (3·99–7·83) |
| CS‡ >1:64 | 68 | 1 625 | 41·82 | 12·63 (9·95–16·02) | 12·73 (10·03–16·15) | 10·55 (8·26–13·49) |
| **Mortality between 1–4 years** | | | | | | |
| Without CS | 26 489 | 56 659 192 | 0·46 | Ref | Ref | Ref |
| CS‡ ≤1:16 | 148 | 155 946 | 0·94 | 1·93 (1·65–2·27) | 1·86 (1·58–2·19) | 1·60 (1·35–1·88) |
| CS‡ >1:16 ≤1:32 | 15 | 9 605 | 1·56 | 3·21 (1·93–5·33) | 3·11 (1·87–5·17) | 2·38 (1·38–4·10) |
| CS‡ >1:32 ≤1:64 | 5 | 4 286 | 1·16 | 2·40 (0·99–5·76) | 2·32 (0·96–5·59) | 1·64 (0·61–4·37) |
| CS‡ >1:64 | 5 | 4 327 | 1·15 | 2·37 (0·98–5·71) | 2·32 (0·96–5·58) | 2·08 (0·86–5·01) |
| **Under-five mortality (under 5 years)** | | | | | | |
| Without CS | 223 483 | 76 403 230 | 2·92 | Ref | Ref | Ref |
| CS‡ ≤1:16 | 1 088 | 220 466 | 4·93 | 1·50 (1·41–1·59) | 1·50 (1·41–1·59) | 1·29 (1·22–1·38) |
| CS‡ >1:16 ≤1:32 | 187 | 13 499 | 13·85 | 4·23 (3·66–4·88) | 4·23 (3·66–4·88) | 3·48 (2·99–4·05) |
| CS‡ >1:32 ≤1:64 | 132 | 6 024 | 21·90 | 6·61 (5·57–7·84) | 6·62 (5·58–7·85) | 5·41 (4·50–6·49) |
| CS‡ >1:64 | 206 | 6 090 | 33·82 | 10·03 (8·75–11·50) | 10·10 (8·81–11·58) | 8·87 (7·70–10·22) |

‡Those individuals with SINAN registry.

*Partly adjusted for region and year of birth.

†Adjusted for maternal age, education, marital status, and self-declared race, region, and year of birth.

CI, confidence interval; CS, congenital syphilis; HR, hazard ratio.

symptoms can appear after the first year of life. It can also reach the second year of life, which is classified as late CS, commonly affecting bones, teeth, and the nervous system [13,24]. Another pathway CS can cause an infant's death is through complications of being born pre-term or with a low birth weight since live births with CS are more likely to be delivered before 37 weeks of gestation and weight lower than 2.5 kg at birth. In our study, these conditions were more common among infants with CS than those without CS, as previously shown in the literature [25].

**Table 5. Mortality risk by age group and symptoms status among singleton live births in the cohort-linked data, Brazil, 2011–2017.**

| Live births | Number of deaths | Person-years | Mortality rate per 1 000 person-years | Crude model HR (95% CI) | Partly adjusted model HR (95% CI)* | Fully adjusted model HR (95% CI)† |
|---|---|---|---|---|---|---|
| **Neonatal mortality (under 28 days)** | | | | | | |
| Without CS | 137 020 | 1 467 358 | 93·37 | Ref | Ref | Ref |
| CS‡ with symptoms | 597 | 683 | 873·33 | 9·13 (8·43–9·90) | 9·07 (8·37–9·83) | 8·46 (7·77–9·21) |
| CS‡ without symptoms | 899 | 6 140 | 146·39 | 1·56 (1·46–1·67) | 1·56 (1·46–1·66) | 1·38 (1·29–1·48) |
| **Postneonatal mortality (28–364 days)** | | | | | | |
| Without CS | 16 922 212 | 59 974 | 3·28 | Ref | Ref | Ref |
| CS‡ with symptoms | 207 | 7 350 | 25·03 | 7·58 (6·61–8·69) | 7·60 (6·63–8·71) | 6·25 (5·40–7·22) |
| CS‡ without symptoms | 563 | 67 551 | 7·39 | 2·24 (2·07–2·44) | 2·25 (2·07–2·44) | 1·81 (1·66–1·98) |
| **Mortality between 1–4 years** | | | | | | |
| Without CS | 26 489 | 56 659 192 | 0·46 | Ref | Ref | Ref |
| CS‡ with symptoms | 26 | 22 349 | 1·00 | 2·40 (1·63–3·53) | 2·33 (1·58–3·42) | 1·96 (1·30–2·95) |
| CS‡ without symptoms | 184 | 202 023 | 0·91 | 1·86 (1·61–2·16) | 1·80 (1·58–2·08) | 1·52 (1·31–1·77) |
| **Under-five mortality (under 5 years)** | | | | | | |
| Without CS | 223 483 | 76 403 230 | 2·92 | Ref | Ref | Ref |
| CS‡ with symptoms | 830 | 31 310 | 26·50 | 8·02 (7·49–8·59) | 7·96 (7·44–8·52) | 7·10 (6·60–7·63) |
| CS‡ without symptoms | 1 646 | 284 343 | 5·78 | 1·78 (1·69–1·87) | 1·77 (1·69–1·86) | 1·52 (1·45–1·60) |

‡Those individuals with SINAN registry.

*Partly adjusted for region and year of birth.

†Adjusted for maternal age, education, marital status, and self-declared race, newborn sex, region, and year of birth.

CI, confidence interval; CS, congenital syphilis; HR, hazard ratio.

It is also important to highlight that CS was more common among live-born of, most socially vulnerable women, particularly those younger, single, black and mixed/brown (*Pardas*) mothers with fewer years of education [26]. It is well known that CS is a marker of quality of care during neonatal care, and our findings reinforce the inequity in this regard. Therefore, public health action to improve timely diagnosis and treatment focusing on socially vulnerable women could reduce mother-to-child transmission and under-five mortality related to CS. Also, this disproportionate distribution of CS cases with a greater burden among socioeconomically disadvantaged groups may have exacerbated the mortality risk from causes not biologically related to CS, as evidenced by the excess mortality due to external and unknown causes of death in older children with CS. In 2017, the Brazilian Ministry of Health has implemented actions to respond to the syphilis situation in the country. Several nationwide strategies have been implemented, such as centralized acquisition and distribution of diagnostic and treatment supplies, tools for disseminating strategic information to help decision-making locally, and research development [27].

Maternal treatment status, the non-treponemal test titers, and the presence of symptoms at birth significantly affected the mortality of children with CS. Similar results for maternal treatment have been described previously in the United States of America [21]. These results emphasize the importance of early detection and timely maternal treatment in preventing offspring death. Previous studies have suggested that higher infant non-treponemal titers have been associated with newborn symptoms [28]. We observed a relationship between non-treponemal titers and CS mortality in the present study. More studies are needed, but this finding can potentially support the clinical application of non-treponemal tests to monitor CS severity.

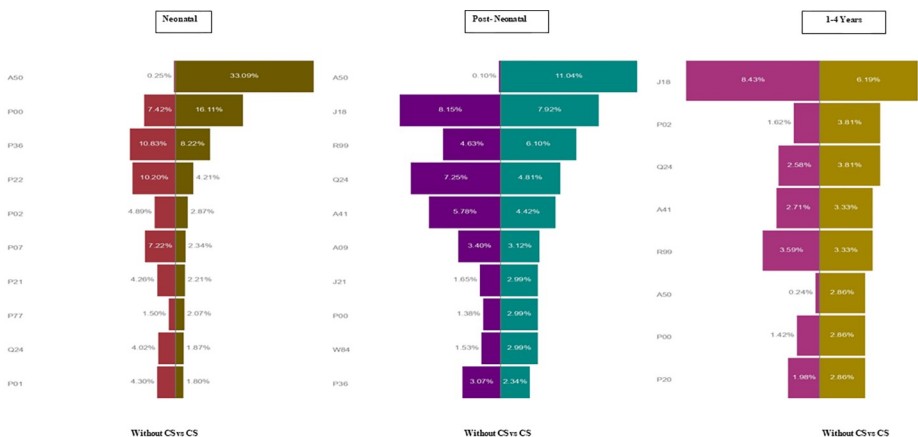

A50: Congenital syphilis P00: Newborn affected by maternal conditions that may be unrelated to present pregnancy P36: Bacterial sepsis of newborn P22: Respiratory distress of newborn P02: Newborn affected by complications of placenta, cord and membranes P07: Disorders of newborn related to short gestation and low birth weight P21: Birth asphyxia P77: Necrotizing enterocolitis of newborn Q24: Other congenital malformations of heart P01: Newborn affected by maternal complications of pregnancy J18: Pneumonia, unspecified organism R99: Ill-defined and unknown cause of mortality A41: Other sepsis A09: Infectious gastroenteritis and colitis, unspecified J21: Acute bronchiolitis W84: Unspecified threat to breathing P20: Intrauterine hypoxia

CS: Congenital syphilis

**Fig 1. Underlying cause of deaths using ICD-10 codes by age group and CS status.** Among children with CS, infections specific to the perinatal period and hematological disorders were more than twice as common as they were among children without CS during the neonatal period. During the postneonatal period, HIV, noxious substances transmitted via placenta or breast milk, anemia, and specific external causes were more than twice as common as they were among children without CS. Among children 1 year or older, bacterial meningitis and paralytic ileus and intestinal obstruction were more than twice as common as they were among children without CS. Similar to the postneonatal period, among older children with CS, specific external causes (events of undetermined intent and accidental threats to breathing) were more than twice as common as they were among children without CS (S2 Fig). CS, congenital syphilis; ICD, International Classification of Diseases.

Around 95% of children registered with CS received some treatment scheme, and most of the children who did not receive treatment died earlier, possibly before a treatment course could be provided. It is also important to highlight that 18% of the live births with CS received a different treatment scheme, which could be related to the penicillin shortage. However, the Brazilian Ministry of Health did not recommend alternative treatment regimens when penicillin was unavailable. We did not analyze the effect of infant treatment on under-five mortality due to susceptibility to selection bias. Frail live births—i.e., children at a high risk of mortality—will be more likely to receive the complete treatment regime with aqueous crystalline penicillin. Therefore, this treatment will be associated with higher mortality risk and biased interpretation of lower effectiveness. The treatment of CS is a very relevant topic and should be better explored when clinical data is available.

A strength of our study was the large sample, which included more than 90 thousand cases of CS. We also had a population-representative comparison group and could control for confounding and analyze the effect of important predictors of CS such as maternal treatment, non-treponemal titers, and presence of symptoms. There are, however, limitations. First, the study was based on registry data, and relevant clinical data were unavailable, including HIV. Furthermore, unobserved confounders can be a source of bias. The diagnosis of CS is complex, and the surveillance system might not capture all the cases; therefore, misclassifying exposure status is possible. We observed an underreport of CS and part of the deaths recorded under the ICD-10 code of CS did not have a SINAN registry. This, in part, can be explained by linkage error. However, we estimated that linkage error would affect 15% of our cases, evidencing underreport of cases in the syphilis information system, as has been evidenced in previous studies [29,30]. This under-registry of CS deaths probably underestimated the mortality risks presented in our study.

Brazil still faces major challenges related to eliminating CS and its adverse outcomes on children's health despite the improvements in access to antenatal care in the last decades, as well as the protocols for diagnosis and treatment for gestational and CS recommended by the Ministry of Health [31]. However, it is necessary to continue the improvements in prenatal care, including screening and treating partners to reduce reinfection [32] test coverage, health worker training about syphilis diagnosis, women's timely treatment, access to penicillin, and follow-up of maternal syphilis and health of the offspring, particularly after the first year. Moreover, the lack of interaction between the surveillance system and care facilities also contributes to rising cases [33].

In summary, this study showed that the excess mortality risk among children with CS goes beyond the first year of life, with the increased risk not totally explained by prematurity and low birth weight status. Our study also highlights the reduced mortality risk associated with maternal treatment, providing evidence of the importance of ensuring timely detection. It is well known that initial and third trimester testing, and early and adequate treatment, are essential in mitigating adverse pregnancy and neonatal outcomes [1,2]. Newborn non-treponemal titers and the presence of signs and symptoms are strong predictors of subsequent mortality. These findings highlight the need for public health action to reduce vertical transmission and CS-related mortality and the importance of instituting well-established postnatal protocols to improve these children's survival.

## Supporting information

**S1 Fig. Flow diagram.**
(PDF)

**S2 Fig. Proportional mortality ratio comparing CS status.**
(PDF)

**S1 Table. RECORD Checklist, the RECORD statement.**
(DOCX)

**S2 Table. Baseline characteristics of deaths.**
(DOCX)

## Author Contributions

**Conceptualization:** Enny S. Paixao, Laura C. Rodrigues.

**Data curation:** Leonardo Salvi, Maria Yury Ichihara.

**Formal analysis:** Enny S. Paixao.

**Funding acquisition:** Enny S. Paixao, Laura C. Rodrigues, Mauricio L. Barreto, Maria Yury Ichihara.

**Investigation:** Eliana Amaral.

**Methodology:** Enny S. Paixao, Rosemeire Fiaccone, Guilherme Lopes de Oliveira.

**Supervision:** Liam Smeeth.

**Validation:** Andrêa JF Ferreira.

**Writing – original draft:** Enny S. Paixao.

**Writing – review & editing:** Enny S. Paixao, Andrêa JF Ferreira, Idália Oliveira dos Santos, Laura C. Rodrigues, Rosemeire Fiaccone, Leonardo Salvi, Guilherme Lopes de Oliveira,

José Guilherme Santana, Andrey Moreira Cardoso, Carlos Antônio de S. S. Teles, Maria Auxiliadora Soares, Eliana Amaral, Liam Smeeth, Mauricio L. Barreto, Maria Yury Ichihara.

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
