## [Editor Report · Decision Letter 0]

10 Oct 2022

Dear Dr Paixao, 

Thank you for submitting your manuscript entitled "Mortality in children under 5 years of age with congenital syphilis: a nationwide cohort study" for consideration by PLOS Medicine.

Your manuscript has now been evaluated by the PLOS Medicine editorial staff as well as by an academic editor with relevant expertise and I am writing to let you know that we would like to send your submission out for external peer review.

Please re-submit your manuscript within two working days, i.e. by Oct 12 2022 11:59PM.

Kind regards,

Philippa Dodd, MBBS MRCP PhD

Editor

PLOS Medicine

---

## [Decision Letter · Decision Letter 1]

12 Jan 2023

Dear Dr. Paixao,

Thank you very much for submitting your manuscript "Mortality in children under 5 years of age with congenital syphilis: a nationwide cohort study" (PMEDICINE-D-22-03293R1) for consideration at PLOS Medicine. 

[LINK]

In light of these reviews, I am afraid that we will not be able to accept the manuscript for publication in the journal in its current form, but we would like to consider a revised version that addresses the reviewers' and editors' comments. Obviously we cannot make any decision about publication until we have seen the revised manuscript and your response, and we plan to seek re-review by one or more of the reviewers. 

We hope to receive your revised manuscript by Feb 02 2023 11:59PM. Please email us (plosmedicine@plos.org) if you have any questions or concerns.

We look forward to receiving your revised manuscript. 

Sincerely,

Callam Davidson

PLOS Medicine

plosmedicine.org

Comments from the Academic Editor:

1) It would help to have a figure showing how the total information from the birth registry yielded the numbers studied in detail. 

2) The cause of death analysis assumes only one ICD code per death, which is a bit odd given misclassification. Could authors provide some information on common combinations by cause? Meningitis in the neonatal period associated with congenital syphilis raises the question of active disease. 

3) Some discussion on the completeness of the birth registry and clustering of these deaths, if any, would be useful.

Please include the study setting (Brazil) in your title. 

In the Abstract Methods and Findings, please include the important covariates adjust for in your analyses.

Please add a final sentence to the Abstract Methods and Findings beginning ‘The main limitations of this study include…’ or similar.

The URL in your Data Availability Statement led to a page that never loaded, please check whether this URL is correct (it may have been an issue with the web page itself).

Please include continuous line numbering throughout your manuscript. 

Please ensure that all numbers in the Abstract match those in the main text. 

Please define all abbreviations in the Author Summary on first use. 

Please briefly describe the study design and include headline numbers (sample size and main results) in your Author Summary.

Please place citations in square brackets, normal script, and preceding punctuation. 

Please check throughout for typos (e.g., ‘titters’ = ‘titres’).

Please confirm whether informed consent was required for this study, and if so, whether this was verbal or written.

I could not locate the RECORD checklist, please include this in your Supporting Information. The checklist should be completed using section headings and paragraph numbers (as opposed to page/line numbers).

Did your study have a prospective protocol or analysis plan? Please state this (either way) early in the Methods section.

Please insert tables into the main manuscript, immediately after the paragraph in which they are first cited.

Please include Table S1 in the main text and rename as Table 1, as this information is central to the manuscript. 

Please define CS in the legend of Figure 1.

Please include the numerator and denominator when stating percentages in the Results. 

Please confirm in the Table S2 legend that figures are n (%) unless otherwise stated.

Given the observational nature of your study, please avoid causal terms like ‘effect’ (paragraph one of the Discussion) and refer instead to associations.

Please temper claims of primacy (paragraph two, Discussion) by including ‘to our knowledge’, or similar.

In your References, please use et al after listing the first six authors.

Please remove and italics formatting in your References.

Comments from the reviewers:

Reviewer #1: Statistical review

This paper reports a population-based cohort study that investigates whether children with congenital syphilis have a higher mortality rate. I had some minor comments, which are provided below:

1. Abstract: I would recommend after the all-cause mortality rate in CS and non-CS groups are given, the overall HR + CI is given (or the crude mortality rate increase given in the third paragraph of the results).

2. Statistical analyses: "we partially adjusted the model" - I did not follow what partial adjustment meant.

3. Statistical analyses: for analyses of mortality aged 1-4 years, I am not sure I followed how events in the first year were dealt with. Were they excluded, or was there a time-varying effect of CS included that allowed the mortality rate to change during different time periods?

4. Results "those with a titter higher than 1:64 was more than 9 … times more likely to die than those without CS" technically the hazard ratio is not the same as the risk ratio, although for small event rates it is approximately equal.

5. Discussion: I agree with the authors's interpretation that linkage error would probably lead to underestimates of association. I wonder if another way in this happens is if CS causes reduction in live birth rate - might this bias the estimates of mortality difference conditional on live birth? 

6. Discussion: I would mention that there is always the chance of unobserved confounders that might cause bias with this study.

James Wason

Reviewer #2: I would like to congratulate the authors for their preparation of such a well-written and timely manuscript, titled "Mortality in children under 5 years of age with congenital syphilis: a nationwide cohort"

The research question is clearly stated and the relevance to contemporary times cannot be overstated. As the authors astutely pointed out, the majority of long-term sequelae in treated neonates remains unknown. The comparison of infection related variables (both maternal and neonatal) makes this one of the best papers I have read regarding the unknowns of future health for congenital syphilis. Adverse health outcomes have been grossly understudied and through this large population analysis, some clarity about the long-term effects have been demonstrated, controlling for known confounders such as prematurity. The results are interesting, and I am surprised to see that even treated newborns had considerable adverse health outcomes. The strengths and limitations are well described.

I highly recommend this manuscript for publication and look forward to seeing it in print. I am thankful for this opportunity to review such a relevant and well written manuscript.

The methods were clearly defined, and I believe were done using guidelines.

The results were clearly demonstrated and were well described in the manuscript. 

I had some questions:

The number of infants dying annually from congenital syphilis is unclear, but estimates suggest over 370 000 perinatal deaths. Do you mean stillbirth and neonatal? Or just neonatal?

In the introduction - you state "globally accepted case definition," Do you mean for the newborn? If you can explain this briefly.

Reviewer #3: Review of Mortality in children under 5 years of age with CS, PlosMed

This is an important analysis demonstrating increased mortality among infants/children with "reported" congenital syphilis. Although it is intuitive that the risk of death would be higher among infants/children with CS, to my knowledge this has not been reported using such a large nationally-derived data set. This analysis fills an important gap and is worthy of publication in PLoSMed.

There is no line numbering or page numbers in the shared draft. There are many wording edits that are needed but it is difficult to describe the placement without line numbering or page numbers. 

Major considerations

1. During 2014-2017, there was a global shortage of penicillin. This had a major impact on the availability of aqueous, procaine and benzathine penicillin in Brazil for treatment of maternal and congenital syphilis. This needs to be clearly stated and referenced. Below are two references but there are other published data on the impact in Brazil. 

a. Nurse-Findlay S, et al. Supply, Demand, and Shortages of Benzathine Penicillin for Treatment of Syphilis: A Market Assessment. PLoS Medicine. 2017;14 (12):e1002473. 

b. Brazil-focused in Rocha AF, Araújo AL et al, Treatment administered to newborns with congenital syphilis during the period of penicillin shortage in Brazil. BMC Pediatr 21, 166 (2021). https://doi.org/10.1186/s12887-021-02619-x

2. Related to item 1, acknowledging the critical impact of shortages of penicillin had on the treatment of maternal and congenital syphilis is an important limitation of the analysis as some of the HR and increased mortality may be associated with the fact that "alternative-non-recommended, non-penicillin" therapies were given to both mothers and infants during the time period of this study. 

3. The authors have access to the treatment regimens given to the infants with CS. Where children that received non-recommended treatment for CS more likely to die? I would recommend that a frequency table be added describing the treatment regimens given to CS infants and that a focused analysis of "non-penicillin regimens" and risk of death be considered

4. Should receipt of a non-penicillin treatment in the mother or the infant be included in the multi-variate model for risk of death? 

5. The authors use a global estimate of 370,000 annual from outdated references 9 and 10 in the Introduction and the Author summary second bullet. Would recommend using the 2016 global estimates from reference number 6 Korenromp et al and not including ref 9 and 10.

6. This reviewer is aware of a period when an additional criteria for CS was "non-treatment of the male partner of the mother" . If true, over counting of CS cases would have occurred during this period diluting the risk of death from CS. The authors can research and confirm or delete. 

7. What is the definition of "abortion". Is this the same thing as "stillbirth". Abortion may be interpreted as an intentional act to "abort" the pregnancy. 

8. For classification of child mortality, "under five mortality" overlaps with the other three categories. Be careful how this is used in the paper

9. In the statistical analyses, you state" covariate included in the adjustments were selected a priori based on the literature. Exactly what are the covariates? Please include these.

10. In the discussion, could bacterial meningitis be neurosyphilis?

11. In the results and limitations would note that this analysis included "reported" cases. Overcounting as well as undercounting may have occurred

12. Would add "access to penicillin" treatment in the list of "improvements" in care in the discussion. 

Minor edits:

1. Please replace "titters" with "titers" throughout

2. In reference to non treponemal tests replace "titles" with "titers"

3. Please replace "reagent" with "reactive" when referring to a "reactive" non-treponemal test

4. Please define "Brazilian territory"

5. Would rephrase "child biopsy" in procedures

6. Throughout the abstract and text there is use of …f or example About 85% or 84.9%. Please use the same method of data presentation throughout. Also for each percent (%), please provide the numerator and denominator of these calculations. 

7. Recommend interpreting the hazard ratios as "risk of death" in the results section 

Reviewer #4: The article "Mortality in children under 5 years of age with congenital syphilis: a nationwide cohort study" deals with a topic of global health interest. Although there is a stigma and prejudice in believing that syphilis is a topic that is more related to developing countries, which is not true, this disease is already a concern in the entire region of the Americas, in China and in countries of the European Union . Therefore, the article deals with a topic of international relevance for public and global health.

Recommendations

[1] The authors cite in the authors' introduction that there is an alarming increase in congenital syphilis in several countries. This is true, as there are several publications that report this problem. Therefore, I recommend that authors include these data in the introduction. Pointing out the countries with these problems is essential to make the relevance of the work clearer, without the reader needing to read its references in order to have this understanding. Of course, it doesn't need to be very detailed (this is in the reference used), but it is important to better qualify the information.

[2] Methodology

Study design We conducted a retrospective population-based cohort study, including all singleton live births in Brazil from January 1, 2011, to December 31, 2017. These live births were followed-up from birth until December 31, 2017, death, or up to five years old.

"including all singleton live births in Brazil from January 1, 2011, to December 31, 2017"

"These live births were followed-up"

How did the authors follow all live births from 2011 to 2017? The authors need to explain this part better.

[2.1]

The information systems used by the authors are those available in Brazil, therefore, it is an official data source, which is very important, as they are data used by decision makers in the area of public health. However, all listed systems have problems and limitations, for example, there is no interoperability between these systems, which is a limitation of the work, so there was a need to use a tool to do the linkage (CIDACS-RL-Record Linkage) which can lead to loss of data and information. Therefore, I recommend authors to better describe the tool used, how it works and if there are already scientific publications that have validated the tool. If there is an article published about the tool, I recommend quoting it, otherwise the authors should describe how the linkage validation tests were carried out on the data used - no matter how good these tools are, they cannot linkage with 100% integrity (this is even mentioned by the authors), similar to models based on relational algebra. What search and similarity algorithms are used in this tool? What is the level of accuracy of these algorithms?. How were exceptions not found handled? How was unrelated data highlighted? What are the limitations of each Information System and how were they treated in the data pre-processing (before linkage)?

[2.2] The authors cite three information systems used, which store sensitive patient data in a non-anonymized manner, therefore, the authors must explicitly inform in the methodology that they were authorized by the Ministry of Health to use this information for research purposes, this will protect the article in relation to possible questions about the General Data Protection Law (LGPD) of Brazil.

[2.3] How many records were processed, to later linkage the data? This information does not appear in the methodology.

[2.4] Notifications of Congenital Syphilis in Brazil are almost not investigated, there is a dissonance between notifications and the investigation of cases for effective confirmation if in fact the child has congenital syphilis. In Brazil today, according to the PCDT, in case of doubt, the notification of congenital syphilis is made, that is, many children may be being notified without actually having congenital syphilis. This subjective criterion included in the most current PCDT and the lack of investigation of cases may be increasing the notification of cases, in addition to the lack of adequate management of cases from prenatal care. How did the authors handle this? In the methodology, they explained which cases were considered, but this issue was not explained.

Results

[3] "We followed 20 057 013 live-born children from birth up to five years", the authors need to better explain how this follow-up was performed.

The results reflect the methodology used, therefore they are described. However, I only reinforce the issue mentioned in this item 3.

Discussion

[4]

The discussion presented by the authors is well qualified, but they could go even further. Discuss weaknesses in the reporting system regarding congenital syphilis - SINAN currently appears to be an obsolete system for providing qualified and timely information about the syphilis epidemic in Brazil - The syphilis bulletin is published only once a year, how to make timely decisions looking back?

In Brazil, syphilis cases are not managed during management until the baby's birth. This is also a reflection of the obsolescence of information systems, the lack of interoperability, and mainly, the lack of integration between the areas of surveillance and health care. This is a point that could also be addressed in the discussion of the article - the authors should evaluate this possibility. Several articles published, more recently, by Brazilian scientists on syphilis deal with this topic.

In Brazil, the recommendation to create vertical transmission investigation committees may be harming the investigation of cases. It should be noted that WHO and PAHO recommend that cases be investigated, but they do not talk about creating a committee for this. Recent surveys report that such committees do not work for the most part, which impairs the qualification of epidemiological data.

Brazil declared a syphilis epidemic in 2016 and since 2017, several actions have been initiated to respond to congenital syphilis, one of which is the Project "No Syphilis", which was developed jointly with the Ministry of Health and PAHO. One of the major contributions of this technical cooperation project was the inclusion of syphilis in the public health agenda of states and municipalities. Considering that the authors made notes about the context of syphilis in Brazil in the discussion, it would be important to go deeper and include the actions that are underway in the country and what has already been developed.

Reviewed by

Ricardo Valentim

Reviewer #5: In this study, the authours aimed to estimate excess all-cause mortality in children under five years with Congenital Syphilis (CS) as compared with those without CS. It was observed an increased mortality risk among children with CS that goes beyond the first year of life. It also reinforces the importance of maternal treatment, that infant non-treponemal titers and the presence of signs and symptoms of CS at birth are strongly associated with subsequent mortality.

This study has an important impact in terms of public health. The findings refer to a representative sample of the population of Brazilian children, observing factors related to the worst clinical outcome of children with CS. The results reinforce that adequate prenatal care and early management of children affected by CS are essential for a good prognosis and lower morbidity and mortality related to the disease. 

The authors acknowledge its limitations, and these limitations do not reduce the importance of the study.

My opinion is in favor of accepting the paper for publication in this journal.

[LINK]

---

## [Decision Letter · Decision Letter 2]

22 Feb 2023

Dear Dr. Paixão,

Thank you very much for re-submitting your manuscript "Mortality in children under 5 years of age with congenital syphilis: a nationwide cohort study in Brazil" (PMEDICINE-D-22-03293R2) for review by PLOS Medicine.

I have discussed the paper with my colleagues and the academic editor and it was also seen again by three reviewers. I am pleased to say that provided the remaining editorial and production issues are dealt with we are planning to accept the paper for publication in the journal.

[LINK]

We look forward to receiving the revised manuscript by Mar 01 2023 11:59PM.   

Sincerely,

Callam Davidson, 

Senior Editor 

PLOS Medicine

plosmedicine.org

Requests from Editors:

Please update your title to “Mortality in children under 5 years of age with congenital syphilis in Brazil: a nationwide cohort study”.

Please correct the typos at lines 262, 272, 311, 319, 353 (‘titters’ ought to be ‘titers’).

As no prospective protocol or analysis plan exists, please make sure that the Methods section transparently describes when analyses were planned, and when/why any data-driven changes to analyses took place. Changes in the analysis should be identified as such in the Methods section of the paper, with rationale.

To help us extend the reach of your research, please provide any Twitter handle(s) that would be appropriate to tag, including your own, your coauthors’, your institution, funder, or lab. Please respond to this email with any handles you wish to be included when we tweet this paper.

Comments from Reviewers:

Reviewer #1: Thank you to the authors for addressing my previous comments. Just one minor thing - I believe that it should be 'hazard' rather than 'hazard ratio' in the change made to my previous comment 4. Otherwise I am happy with the changes and have no further issues to raise.

Reviewer #3: The revisions and comments from other reviewers have improved this manuscript.

I support publication of this revised draft.

Reviewer #4: The authors met all the requested recommendations, congratulations on the research.

[LINK]

---

## [Editor Report · Decision Letter 3]

28 Feb 2023

Dear Dr Paixão, 

On behalf of my colleagues and the Academic Editor, Dr Zulfiqar A. Bhutta, I am pleased to inform you that we have agreed to publish your manuscript "Mortality in children under 5 years of age with congenital syphilis in Brazil: a nationwide cohort study" (PMEDICINE-D-22-03293R3) in PLOS Medicine.

When making the formatting changes, please also address the following editorial requests:

- Please define the abbreviation HR in the abstract on first use.

- Please simplify bullet point three under question two ('What did the researchers do and find?') of the Author Summary. The current wording is difficult to follow. 

- Line 271: Please change 'hazard ratio' to 'hazard'.

PRESS

Sincerely, 

Callam Davidson 

Associate Editor 

PLOS Medicine